# AN OUT-OF-THE-BOX FULL-NETWORK EMBEDDING FOR CONVOLUTIONAL NEURAL NETWORKS

## ABSTRACT

Transfer learning for feature extraction can be used to exploit deep representations in contexts where there is very few training data, where there are limited computational resources, or when tuning the hyper-parameters needed for training is not an option. While previous contributions to feature extraction propose embeddings based on a single layer of the network, in this paper we propose a full-network embedding which successfully integrates convolutional and fully connected features, coming from all layers of a deep convolutional neural network. To do so, the embedding normalizes features in the context of the problem, and discretizes their values to reduce noise and regularize the embedding space. Significantly, this also reduces the computational cost of processing the resultant representations. The proposed method is shown to outperform single layer embeddings on several image classification tasks, while also being more robust to the choice of the pre-trained model used for obtaining the initial features. The performance gap in classification accuracy between thoroughly tuned solutions and the full-network embedding is also reduced, which makes of the proposed approach a competitive solution for a large set of applications.

## 1 INTRODUCTION

Deep learning models, and particularly convolutional neural networks (CNN), have become the standard approach for tackling image processing tasks. The key to the success of these methods lies in the rich representations deep models build, which are generated after an exhaustive and computationally expensive learning process (LeCun et al., 2015). To generate deep representations, deep learning models have strong training requirements in terms of dataset size, computational power and optimal hyper-parametrization. For any domain or application in which either of those factors is an issue, training a deep model from scratch becomes unfeasible.

Within deep learning, the field of transfer learning studies how to extract and reuse pre-trained deep representations. This approach has three main applications: improving the performance of a network by initializing its training from a non-random state (Xu et al., 2015; Branson et al., 2014; Liu et al., 2016), enabling the training of deep networks for tasks of limited dataset size (Ge & Yu, 2017; Simon & Rodner, 2015), and exploiting deep representations through alternative machine learning methods (Azizpour et al., 2016; Sharif Razavian et al., 2014; Gong et al., 2014). The first two cases, where training a deep network remains the end purpose of the transfer learning process, are commonly known as *transfer learning for fine-tuning*, while the third case, where the end purpose of the transfer learning does not necessarily include training a deep net, is typically referred as *transfer learning for feature extraction*.

Of the three limiting factors of training deep networks (*i.e.*, dataset size, computational cost, and optimal hyper-parametrization), transfer learning for fine-tuning partly solves the first. Indeed, one can successfully train a CNN on a dataset composed by roughly a few thousand instances using a pre-trained model as starting point, and achieve state-of-the-art-results. Unfortunately, fine-tuning a model still requires a minimum dataset size, a significant amount of computational resources, and lots of time to optimize the multiple hyper-parameters involved in the process.

Transfer learning for feature extraction on the other hand is based on processing a set of data instances through a pre-trained neural network, extracting the activation values so these can be used by another learning mechanism. This is applicable to datasets of any size, as each data instance

is processed independently. It has a relatively small computational cost, since there is no deep net training. And finally, it requires no hyper-parameter optimization, since the pre-trained model can be used out-of-the-box. Significantly, the applications of transfer learning for feature extraction are limited only by the capabilities of the methods that one can execute on top of the generated deep representations.

As previously mentioned, designing and training a deep model to maximize classification performance is a time consuming task. In this paper we explore the opposite approach, minimizing the design and tuning effort using a feature extraction process. Our goal is to build an out-of-the-box classification tool (which could be used by anyone regardless of technical background) capable of defining a full-network embedding (integrating the representations built by all layers of a source CNN model). When compared to single-layer embeddings, this approach generates richer and more powerful embeddings, while also being more robust to the use of inappropriate pre-trained models. We asses the performance of such solution when compared with thoroughly designed and tuned models.

## 2  RELATED WORK

Transfer learning studies how to extract and reuse deep representations learnt for a given task $t0$, to solve a different task $t1$. Fine tuning approaches require the $t1$ target dataset to be composed by at least a few thousands instances, to avoid overfitting during the fine-tuning process. To mitigate this limitation, it has been proposed to reuse carefully selected parts of the $t0$ dataset in the fine-tuning process alongside the $t1$ (*i.e.*, selective joint fine-tuning) (Ge & Yu, 2017), and also to use large amounts of noisy web imagery alongside with clean curated data (Krause et al., 2016). In fine-tuning, choosing which layers of weights from the $t0$ model should be transferred, and which should be transferred and kept unchanged on the $t1$ training phase has a large impact on performance. Extensive research on that regard has shown that the optimal policy depends mostly on the properties of both $t0$ and $t1$ (Azizpour et al., 2016; Yosinski et al., 2014; Long et al., 2015). This dependency, together with the hyper-parameters inherent to deep network training, defines a large set of problem specific adaptations to be done by fine-tuning solutions.

Given a pre-trained model for $t0$ one may use alternative machine learning methods for solving $t1$, instead of fine-tuning a deep model. For that purpose, one needs to generate a representation of $t1$ data instances as perceived by the model trained for $t0$. This feature extraction process is done through a forward pass of $t1$ data instances on the pre-trained CNN model, which defines a data embedding that can be fed to the chosen machine learning method (*e.g.*, a Support Vector Machine, or SVM, for classification). In most cases, the embedding is defined by capturing and storing the activation values of a single layer close to the output (Azizpour et al., 2016; Sharif Razavian et al., 2014; Gong et al., 2014; Donahue et al., 2014; Mousavian & Kosecka, 2015; Ren et al., 2017). The rest of layers (*e.g.*, most convolutional layers) are discarded because these "are unlikely to contain a richer semantic representation than the later feature" (Donahue et al., 2014). So far, this choice has been supported by performance comparisons of single-layer embeddings, where high-level layer embeddings have been shown to consistently outperform low-level layer embeddings (Azizpour et al., 2016; Sharif Razavian et al., 2014). However, it is also known that all layers within a deep network, including low-level ones, can contribute to the characterization of the data in different ways (Garcia-Gasulla et al., 2017). This implies that the richest and most versatile representation that can be generated by a feature extraction process must include all layers from the network, *i.e.*, it must define a full-network embedding. However, no full-network embedding has been proposed in the literature so far, due to the difficulty of successfully integrating the features found on such an heterogeneous set of layers as the one defined by a full deep network architecture.

Beyond the layers to extract, there are many other parameters that can affect the feature extraction process. Some of those are evaluated in (Azizpour et al., 2016), which includes parameters related with the architecture and training of the initial CNN (*e.g.*, network depth and width, distribution of training data, optimization parameters), and parameters related with transfer learning process (*e.g.*, fine-tuning, spatial pooling and dimensionality reduction). Among the most well established transformations of deep embeddings are a L2 normalization (Azizpour et al., 2016; Sharif Razavian et al., 2014), and an unsupervised feature reduction like Principal Component Analysis (PCA) (Azizpour et al., 2016; Sharif Razavian et al., 2014; Gong et al., 2014). The quality of the resultant embedding

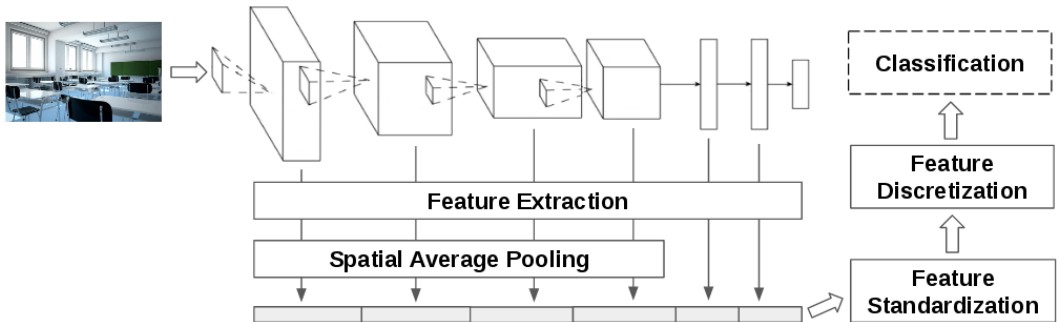

Figure 1: Overview of the proposed out-of-the-box full-network embedding generation workflow.

is typically evaluated by the performance of an SVM, trained using the embedding representations, to solve a classification task (Azizpour et al., 2016; Sharif Razavian et al., 2014). Recently, extracted features have also been combined with more sophisticated computer vision techniques, such as the constellation model (Simon & Rodner, 2015) and Fisher vectors (Cimpoi et al., 2015), with significant success.

## 3    FULL-NETWORK EMBEDDING

Transfer learning for feature extraction is used to embed a dataset $t1$ in the representation language learnt for a task $t0$. To do so, one must forward pass each data instance of $t1$ through the model pre-trained on $t0$, capturing the internal activations of the net. This is the *first step* of our method, but, unlike previous contributions to feature extraction, we store the activation values generated on every convolutional and fully connected layer of the model to generate a full-network embedding.

Each filter within a convolutional layer generates several activations for a given input, as a result of convolving the filter. This corresponds to the presence of the filter at various locations of the input. In a resultant feature extracted embedding this implies a significant increase in dimensionality, which is in most cases counterproductive. At the same time, the several values generated by a given filter provide only relative spatial information, which may not be particularly relevant in a transfer learning setting (*i.e.*, one where the problem for which the filters were learnt is not the same as the problem where the filter is applied). To tackle this issue, a recurrent solution in the field is to perform a *spatial average pooling* on each convolutional filter, such that a single value per filter is obtained by averaging all its spatially-depending activations (Azizpour et al., 2016; Sharif Razavian et al., 2014). After this pooling operation, each feature in the embedding corresponds to the degree with which a convolutional filter is found on average in the whole image, regardless of location. While losing spatial information, this solution maintains most of the embedding descriptive power, as all convolutional filters remain represented. A spatial average pooling on the filters of convolutional layers is the *second step* of our method. The values resulting from the spatial pooling are concatenated with the features from the fully connected layers into a single vector, to generate a complete embedding. In the case of the well-known `VGG16` architecture (Simonyan & Zisserman, 2014) this embedding vector is composed by 12,416 features.

The features composing the embedding vector so far described are obtained from neurons of different type (*e.g.*, convolutional and fully connected layers) and location (*i.e.*, from any layer depth). These differences account for large variations in the corresponding feature activations (*e.g.*, distribution, magnitude, *etc.*). Since our method considers an heterogeneous set of features, a *feature standardization* is needed. Our proposed standardization computes the z-values of each feature, using the train set to compute the mean and standard deviation. This process transforms each feature value so that it indicates how separated the value is from the feature mean in terms of positive/negative standard deviations. In other words, the degree with which the feature value is atypically high (if positive) or atypically low (if negative) in the context of the dataset. A similar type of feature normalization is frequently used in deep network training (*i.e.*, batch normalization) (Ioffe & Szegedy, 2015), but this is the first time this technique has been applied to a feature extraction solution. As discussed in §2, most feature extraction approaches apply an L2 norm by data instance, thus nor-

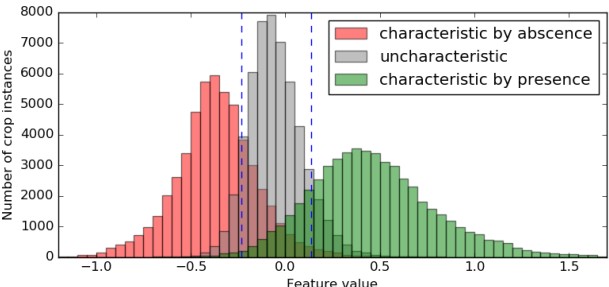

Figure 2: For the *mit67* dataset, distribution of average standardized feature values for those features belonging to the sets identified in (Garcia-Gasulla et al., 2017). Vertical dashed lines mark the $ft^-$ and $ft^+$ thresholds separating the two pairs of distributions as computed by the Kolmogrov-Smirnov statistic.

Table 1: Feature value thresholds $ft^+$ and $ft^-$ found by computing the Kolmogrov-Smirnov statistic of the distributions exemplified in Figure 2.

| Dataset | $ft^+$ | $ft^-$ |
|---------|--------|--------|
| mit67 | 0.14 | -0.23 |
| cub200 | 0.20 | -0.24 |
| flowers102 | 0.15 | -0.24 |

malizing by image instead of by feature. As seen in §5, this approach provides competitive results, but is not appropriate when using features coming from many different layers. By using the z-values per feature, we use activations across the dataset as reference for normalization. This balances each feature individually, which allows us to successfully integrate all types of features in the embedding. Significantly, this feature standardization process generates a context dependent embedding, as the representation of each instance depends on the rest of instances being computed with it. Indeed, consider how the features relevant for characterizing a bird in a context of cars are different than the ones relevant for characterizing the same bird in a context of other birds. Such a subjective representation makes the approach more versatile, as it is inherently customized for each specific problem. After the feature standardization, the final step of the proposed pipeline is a feature discretization, which is described in §3.1. An end-to-end overview of the proposed embedding generation is shown in Figure 1.

## 3.1 Feature Discretization

The embedding vectors we generate are composed of a large number of features. Exploring a representation space of such high-dimensionality is problematic for most machine learning algorithms, as it can lead to overfitting and other issues related with the curse of dimensionality. A common solution is to use dimensionality reduction techniques like PCA (Mousavian & Kosecka, 2015; Azizpour et al., 2016; Carvalho et al., 2016). We propose an alternative approach, which keeps the same number of features (and thus keeps the size of the representation language defined by the embedding) but reduces their expressiveness similarly to quantization methodology followed in (Carvalho et al., 2016). In detail, we discretize each standardized feature value to represent either an atypically low value (-1), a typical value (0), or an atypically high value (1). This discretization is done by mapping feature values to the $\{-1, 0, 1\}$ domain by defining two thresholds $ft^-$ and $ft^+$.

To find consistent thresholds, we consider the work of Garcia-Gasulla et al. (2017), who use a supervised statistical approach to evaluate the importance of CNN features for characterization. Given a feature $f$ and a class $c$, this work uses an empirical statistic to measure the difference between activation values of $f$ for instances of $c$ and activation values of $f$ for the rest of classes in the dataset. This allows them to quantify the relevance of feature/class pairs for class characterization. In their work, authors further separate these pairs in three different sets: *characteristic by abscence*, *uncharacteristic* and *characteristic by presence*. We use these three sets to find our thresholds $ft^-$ and $ft^+$, by mapping the feature/class relevances to our corresponding feature/image activations. We do so on some datasets explored in (Garcia-Gasulla et al., 2017): *mit67*, *flowers102* and *cub200*, by computing the average values of the features belonging to each of the three sets. Figure 2 shows the three resulting distributions of values for the *mit67* dataset. Clearly, a strong correlation exists between the supervised statistic feature relevance defined by Garcia-Gasulla et al. (2017) and the standardized feature values generated by the full-network embedding, as features in the characteristic by absence set correspond to activations which are particularly low, while features in the

Table 2: Properties of all datasets computed

| Dataset | #Images | #Classes | #Images (train) | #Images (test) | #Images per class | #Images per class (train) | #Images per class (test) |
|---|---|---|---|---|---|---|---|
| mit67 | 6,700 | 67 | 5,360 | 1,340 | 100 | 77 - 83 | 17 - 23 |
| cub200 | 11,788 | 200 | 5,994 | 5,794 | 41 - 60 | 29 - 30 | 12 - 30 |
| flowers102 | 8,189 | 102 | 2,040 | 6,149 | 40 - 258 | 20 | 20 - 238 |
| cats-dogs | 7,349 | 37 | 3,680 | 3,669 | 184 - 200 | 93 - 100 | 88 - 100 |
| sdogs | 20,580 | 120 | 12,000 | 8,580 | 150 - 200 | 100 | 50 - 100 |
| caltech101 | 9,146 | 101 | 3,060 | 2,995 | 31 - 800 | 30 | 1 - 50 |
| food101 | 25,250 | 101 | 20,200 | 5,050 | 250 | 200 | 50 |
| textures | 5,640 | 47 | 3,760 | 1,880 | 120 | 80 | 40 |
| wood | 438 | 7 | 350 | 88 | 14 - 179 | 10 - 142 | 3 - 37 |

characteristic by presence set correspond to activations which are particularly high. We obtain the $ft^-$ and $ft^+$ values through the Kolmogrov-Smirnov statistic, which provides the maximum gap between two empirical distributions. Vertical dashed lines of Figure 2 indicate these optimal thresholds for the *mit67* dataset, the rest are shown in Table 1. To obtain a parameter free methodology, and considering the stable behavior of the $ft^+$ and $ft^-$ thresholds, we chose to set $ft^+ = 0.15$ and $ft^- = -0.25$ in all our experiments. Thus, after the step of feature standardization, we discretize the values above $0.15$ to $1$, the values below $-0.25$ to $-1$, and the rest to $0$.

## 4 DATASETS

One of the goals of this paper is to identify a full-network feature extraction methodology which provides competitive results out-of-the-box. For that purpose, we evaluate the embedding proposed in §3.1 on a set of 9 datasets which define different image classification challenges. The list includes datasets of classic object categorization, fine-grained categorization, and scene and textures classification. The disparate type of discriminative features needed to solve each of these problems represents a challenge for any approach which tries to solve them without specific tuning of any kind.

The MIT Indoor Scene Recognition dataset (Quattoni & Torralba, 2009) (*mit67*) consists of different indoor scenes to be classified in 67 categories. Its main challenge resides in the class dependence on global spatial properties and on the relative presence of objects. The Caltech-UCSD Birds-200-2011 dataset (Wah et al., 2011) (*cub200*) is a fine-grained dataset containing images of 200 different species of birds. The Oxford Flower dataset (Nilsback & Zisserman, 2008) (*flowers102*) is a fine-grained dataset consisting of 102 flower categories. The dataset contains only 20 samples per class for training. The Oxford-IIIT-Pet dataset (Parkhi et al., 2012) (*cats-dogs*) is a fine-grained dataset covering 37 different breeds of cats and dogs. The Stanford Dogs dataset (Khosla et al., 2011) (*sdogs*) contains images from the 120 breeds of dogs found in ImageNet. The dataset is complicated by little inter-class variation, and large intra-class and background variation. The Caltech 101 dataset (Fei-Fei et al., 2007) (*caltech101*) is a classical dataset of 101 object categories containing clean images with low level of occlusion. The Food-101 dataset (Bossard et al., 2014) (*food101*) is a large dataset of 101 food categories. Test labels are reliable but train images are noisy (*e.g.*, occasionally mislabeled). The Describable Textures Dataset (Cimpoi et al., 2014) (*textures*) is a database of textures categorized according to a list of 47 terms inspired from human perception. The Oulu Knots dataset (Silvén et al., 2003) (*wood*) contains knot images from spruce wood, classified according to Nordic Standards. This dataset of industrial application is considered to be challenging even for human experts.

Details for these datasets are provided in Table 2. This includes the train/test splits used in our experiments. In most cases we follow the train/test splits as provided by the dataset authors in order to obtain comparable results. A specific case is *caltech101* where, following the dataset authors in-

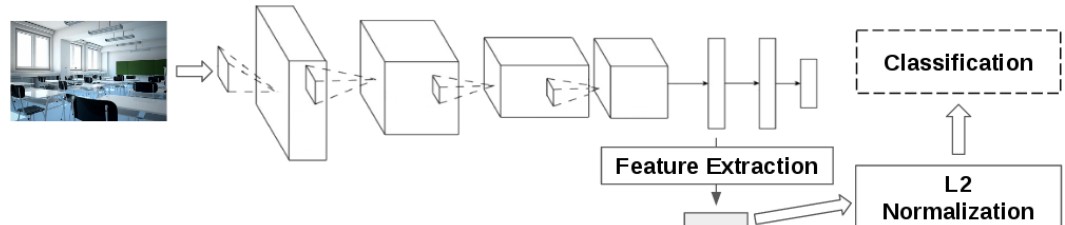

Figure 3: Overview of the baseline embedding generation workflow.

structions (Fei-Fei et al., 2007), we randomly choose 30 training examples per class and a maximum of 50 for test, and repeat this experiment 5 times. The other particular case is the *food101* dataset. Due to its large size, we use only the provided test set for both training and testing, using a stratified 5-fold cross validation. The same stratified 5-fold cross validation approach is used for the *wood* dataset, where no split is provided by the authors.

## 5 CLASSIFICATION EXPERIMENTS

In this section we analyze the performance gap between thoroughly tuned models (those which currently provide state-of-the-art results) and the approach described in §3. To evaluate the consistency of our method out-of-the-box, we decide *not* to use additional data when available on the dataset (*e.g.*, image segmentation, regions of interest or other metadata), or to perform any other type of problem specific adaptation (*e.g.*, tuning hyper-parameters).

As source model for the feature extraction process we use the classical VGG16 CNN architecture (Simonyan & Zisserman, 2014) pre-trained on the Places2 scene recognition dataset (Zhou et al., 2016) for the *mit67* experiments, and the same VGG16 architecture pre-trained on the *ImageNet 2012* classification dataset (Russakovsky et al., 2015) for the rest (these define our $t0$ tasks). As a result the proposed embedding is composed by 12,416. On top of that, we use a linear SVM with the default hyperparameter $C = 1$ for classification, with a one-vs-the-rest strategy. Standard data augmentation is used in the SVM training, using 5 crops per sample (4 corners + central) with horizontal mirroring (total of 10 crops per sample). At test time, all the 10 crops are classified, using a voting strategy to decide the label of each data sample.

Beyond the comparison with the current state-of-the-art, we also compare our approach with the most frequently used feature extraction solution. As discussed in §2, a popular embedding is obtained by extracting the activations of one of the fully connected layers (`fc6` or `fc7` for the VGG16 model) and applying a L2 normalization per data instance (Azizpour et al., 2016; Sharif Razavian et al., 2014; Donahue et al., 2014). We call this our *baseline* method, an overview of it is shown in Figure 3. The same pre-trained model used as source for the full-network embedding is used for the baseline. For both baselines (`fc6` and `fc7`), the final embedding is composed by 4,096 features. This is used to train the same type of SVM classifier trained with the full-network embedding.

### 5.1 CLASSIFICATION RESULTS

The results of our classification experiments are shown in Table 3. Performance is measured with average per-class classification accuracy. For each dataset we provide the accuracy provided by the baselines, by our method, and by the best method we found in the literature (*i.e.*, the state-of-the-art or SotA). For a proper interpretation of the performance gap between the SotA methods and ours, we further indicate if the SotA uses external data (beyond the $t1$ dataset and the $t0$ model) and if it performs fine-tuning.

Overall, our method outperforms the best baseline (`fc6`) by 2.2% accuracy on average. This indicates that the proposed full-network embedding successfully integrates the representations generated at the various layers. The datasets where the baseline performs similarly or slightly outperforms the full-network embedding (*cub200*, *cats-dogs* and *sdogs*) are those where the target task $t1$ overlaps with the source task $t0$ (*e.g.*, *ImageNet 2012*). The largest difference happens for the *sdogs*, which is explicitly a subset of *ImageNet 2012*. In this sort of *easy* transfer learning problems, the fully con-

Table 3: Classification results in % of average per-class accuracy for the baselines, for the full-network embedding, and for the current state-of-the-art (SotA). ED: SotA uses external data, FT: SotA performs fine-tuning of the network. SotA citation for each dataset: *mit67* (Ge & Yu, 2017), *cub200* (Krause et al., 2016), *flowers102* (Ge & Yu, 2017), *cats-dogs* (Simon & Rodner, 2015), *sdogs* (Ge & Yu, 2017), *caltech101* (He et al., 2014), *food101* (Liu et al., 2016) and *textures* (Cimpoi et al., 2015).

| Dataset | mit67 | cub200 | flowers102 | cats-dogs | sdogs | caltech101 | food101 | textures | wood |
|---|---|---|---|---|---|---|---|---|---|
| Baseline fc6 | 80.0 | 65.8 | 89.5 | 89.3 | 78.0 | $91.4_{\pm 0.6}$ | $61.4_{\pm 0.2}$ | 69.6 | $70.8_{\pm 6.6}$ |
| Baseline fc7 | 81.7 | 63.2 | 87.0 | 89.6 | 79.3 | $89.7_{\pm 0.3}$ | $59.1_{\pm 0.6}$ | 69.0 | $68.9_{\pm 6.8}$ |
| Full-network | 83.6 | 65.5 | 93.3 | 89.2 | 78.8 | $91.4_{\pm 0.6}$ | $67.0_{\pm 0.7}$ | 73.0 | $74.1_{\pm 6.9}$ |
| SotA | 86.9 | 92.3 | 97.0 | 91.6 | 90.3 | 93.4 | 77.4 | 75.5 | - |
| ED | ✓ | ✓ | ✓ | ✗ | ✓ | ✗ | ✗ | ✗ | - |
| FT | ✓ | ✓ | ✓ | ✓ | ✓ | ✓ | ✓ | ✗ | - |

nected layer used by the baseline methods has been partly optimized to solve the t1 problem during the original CNN training phase. This explains why the baselines based on the fully connected layers are particularly competitive on these datasets.

State-of-the-art performance is in most cases a few accuracy points above the performance of the full-network embedding (7.8% accuracy on average). These results are encouraging, considering that our method uses no additional data, requires no tuning of parameters and it is computationally cheap (*e.g.*, it does not require deep network training). The dataset where our full-network embedding is more clearly outperformed is the *cub200*. In this dataset Krause et al. (2016) achieve a remarkable state-of-the-art performance by using lots of additional data (roughly 5 million additional images of birds) to train a deep network from scratch, and then fine-tune the model using the *cub200* dataset. In this case, the large gap in performance is caused by the huge disparity in the amount of training data used. A similar context happens in the evaluation of *food101*, where Liu et al. (2016) use the complete training set for fine-tuning, while we only use a subset of the test set (see §4 for details). If we consider the results for the other 6 datasets, the average performance gap between the state-of-the-art and the full-network embedding is 4.2% accuracy on average.

Among the methods which achieve the best performance on at least one dataset, there is one which is not based on fine tuning. The work of (Cimpoi et al., 2015) obtains the best results for the *textures* dataset by using a combination of bag-of-visual-words, Fisher vectors and convolutional filters. Authors demonstrate how this approach is particularly competitive on texture based datasets. Our more generic methodology is capable of obtaining an accuracy 2.5% accuracy lower in this highly specific domain.

The *wood* dataset is designed to be particularly challenging, even for human experts; according to the dataset authors the global accuracy of an experienced human sorter is about 75-85% (Lampinen & Smolander, 1994; Silvén et al., 2003). There is currently no reported results in average per-class accuracy for this dataset, so the corresponding values in Table 3 are left blank. Consequently, the results we report represent the current state-of-the-art to the best of our knowledge ($74.1\%_{\pm 6.9}$ in average per-class accuracy). The best results previously reported in the literature for *wood* correspond to Ren et al. (2017), which are 94.3% in global accuracy. However, the difference between average per-class accuracy and global accuracy is particularly relevant in this dataset, given the variance in images per class (from 3 to 37). To evaluate the average per-class accuracy, we tried our best to replicate the method of Ren et al. (2017), which resulted in $71.0\%_{\pm 8.2}$ average per-class accuracy when doing a stratified 5-fold cross validation. A performance similar to that of our baseline method.

Table 4: Mean per-class accuracy in % obtained by the full-network embedding, when not performing feature discretization (FS), and when only discretizing the values between thresholds ($\{-v, 0, v\}$).

| Dataset | mit67 | cub200 | flowers102 | cats-dogs | sdogs | caltech101 | food101 | textures | wood |
|---|---|---|---|---|---|---|---|---|---|
| FS | 81.0 | 64.9 | 94.1 | 89.9 | 77.3 | 91.5 | 65.3 | 69.6 | 71.5 |
| $\{-v, 0, v\}$ | 82.0 | 64.8 | 93.2 | 89.3 | 77.4 | 92.0 | 66.6 | 70.1 | 70.6 |
| Full-network | 83.6 | 65.5 | 93.3 | 89.2 | 78.8 | 91.5 | 67.0 | 73.0 | 74.1 |

## 5.2 STUDY OF VARIANTS

In this section we consider removing and altering some of the components of the full-network embedding to understand their impact. First we remove feature discretization, and evaluate the embeddings obtained after the feature standardization step (FS). Secondly, we consider a partial feature discretization which only maps values between $ft^+$ and $ft^-$ to zero, and evaluate an embedding which keeps the rest of the original values ($\{-v, 0, v\}$). The purpose of this second experiment is to study if the increase in performance provided by the feature discretization is caused by the noise reduction effect of mapping frequent values to 0, or if it is caused by the space simplification resultant of mapping all activations to only three values.

As shown in Table 4, the full-network embedding outperforms all the other variants, with the exceptions of *flowers102* and *cats-dogs* where FS is slightly more competitive (0.8,0.7% accuracy) and *caltech101* where the best is $\{-v, 0, v\}$ by 0.5% accuracy . The noise reduction variant (*i.e.*, $\{-v, 0, v\}$) outperforms the FS variant in 5 out of 9 datasets. The main difference between both is that the former sparsifies the embeddings by transforming *typical* values to zeros, with few informative data being lost in the process. The complete feature discretization done by the full-network model (*i.e.*, $\{-1, 0, 1\}$) further boosts performance, outperforming the $\{-v, 0, v\}$ embedding on 7 of 9 datasets. This shows the potential benefit of reducing the complexity of the embedding space.

The feature discretization also has the desirable effect of reducing the training cost of the SVM applied on the resulting embedding. Using the FS embedding as control (the slowest of all variants), the $\{-v, 0, v\}$ embedding trains the SVM between 3 and 13 times faster depending on the dataset, while the full-network embedding with its complete discretization trains between 10 and 50 times faster. Significantly, all three embeddings are composed by 12,416 features. For comparison, the baseline method, which uses shorter embeddings of 4,096 features, trains the SVM between 100 and 650 times faster than the FS. For both the baseline and the full-network embeddings, training the SVM takes a few minutes on a single CPU.

A different variation we consider is to an inappropriate task t0 as source for generating the baseline and full-network embeddings. This tests the robustness of each embedding when using an ill-suited pre-trained model. We use the model pre-trained on *ImageNet 2012* for generating the *mit67* embeddings, and the model pre-trained on *Places2* for the rest of datasets. Table 5 shows that the full-network embedding is much more robust, with an average reduction in accuracy of 16.4%, against 24.6% of the baseline. This results remark the limitation of the baseline caused by its own late layer dependency. Finally, we also considered using different network depths, a parameter also analyzed in (Azizpour et al., 2016). We repeated the full-network experiments using the VGG19 architecture instead of the VGG16, and found performance differences to be minimal (maximum difference of 0.3%) and inconsistent.

## 6 CONCLUSIONS

In this paper we describe a feature extraction process which leverages the information encoded in all the features of a deep CNN. The full-network embedding introduces the use of feature standardization and of a novel feature discretization methodology. The former provides context-dependent

Table 5: Classification results in % average per-class accuracy of the baseline and the full-network embedding when using a network pre-trained on *ImageNet 2012* for *mit67* and on *Places2* for the rest.

| Dataset | mit67 | cub200 | flowers102 | cats-dogs | sdogs | caltech101 | food101 | textures | wood |
|---|---|---|---|---|---|---|---|---|---|
| Baseline fc7 | 72.2 | 23.6 | 73.3 | 38.7 | 24.7 | 72.0 | 40.5 | 55.8 | 65.3 |
| Full-network | 75.5 | 35.5 | 88.7 | 56.2 | 37.8 | 80.0 | 55.9 | 65.1 | 74.0 |

embeddings, which adapt the representations to the problem at hand. The later reduces noise and regularizes the embedding space while keeping the size of the original representation language (*i.e.*, the pre-trained model used as source). Significantly, the feature discretization restricts the computational overhead resultant of processing much larger embeddings when training an SVM. Our experiments also show that the full-network is more robust than single-layer embeddings when an appropriate source model is not available.

The resultant full-network embedding is shown to outperform single-layer embeddings in several classification tasks, and to provide the best reported results on one of those tasks (*wood*). Within the state-of-the-art, the full-network embedding represents the best available solution when one of the following conditions apply: When the accessible data is scarce, or an appropriate pre-trained model is not available (*e.g.*, specialized industrial applications), when computational resources are limited (*e.g.*, no GPUs availability), or when development time or technical expertise is restricted or non cost-effective.

Beyond classification, the full-network embedding may be of relevance for any task exploiting visual embeddings. For example, in image retrieval and image annotation tasks, the full-network embedding has been shown to provide a boost in performance when compared to one layer embeddings (Vilalta et al., 2017).

ACKNOWLEDGMENTS

This work is partially supported by the Joint Study Agreement no. W156463 under the IBM/BSC Deep Learning Center agreement, by the Spanish Government through Programa Severo Ochoa (SEV-2015-0493), by the Spanish Ministry of Science and Technology through TIN2015-65316-P project and by the Generalitat de Catalunya (contracts 2014-SGR-1051), and by the Core Research for Evolutional Science and Technology (CREST) program of Japan Science and Technology Agency (JST).

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
