# OpenReview forum: "An Out-of-the-box Full-network Embedding for Convolutional Neural Networks"
_ICLR.cc/2018/Conference — Reject_

### Official Review · AnonReviewer3 · 2017-11-28
**Lack of novelty**

**Rating:** 3
**Confidence:** 4

**Review:**

This paper proposes an out-of-the-box embedding for image classification task. Instead of taking one single layer output from pre-trained network as the feature vector for new dataset, the method first extracts the activations from all the layers, then runs spatial average pooling on all convolutional layers, then normalizes the feature and uses two predefined thresholds to discretize the features to {-1, 0, 1}. Final prediction is learned through a SVM model using those embeddings. Experimental results on nine different datasets show that this embedding outperforms baseline of using one single layer. I think in general this paper lacks novelty and it shouldn't be surprising that activations from all layers should be more representative than one single layer representation. Moreover, in Table 4, it shows that discretization actually hurts the performance. It is also very heuristic to choose the two thresholds.

---

### Official Review · AnonReviewer1 · 2017-11-30
**has novelty issue, results are not impressive**

**Rating:** 4
**Confidence:** 5

**Review:**

The paper addresses the scenario when using a pretrained deep network as learnt feature representation for another (small) task where retraining is not an option or not desired. In this situation it proposes to use all layers of the network to extract feature from, instead of only one layer.
Then it proposes to standardize different dimensions of the features based on their response on the original task. Finally, it discretize each dimension into {-1, 0, 1} to compress the final concatenated feature representation.
Doing this, it shows improvements over using a single layer for 9 target image classification datasets including object, scene, texture, material, and animals.

The reviewer does not find the paper suitable for publication at ICLR due to the following reasons:
- The paper is incremental with limited novelty.
- the results are not encouraging
- the pipeline of standardization, discretization is relatively costly, the final feature vector still large.
- combining different layers, as the only contribution of the paper, has been done in the literature before,  for instance:
“The Treasure beneath Convolutional Layers: Cross-convolutional-layer Pooling
for Image Classification” CVPR 2016

---

### Official Review · AnonReviewer2 · 2017-12-11
**Poor presentation and lack of novelty**

**Rating:** 4
**Confidence:** 5

**Review:**

Paper claims to propose a deep transfer learning method. There are several reasons not to consider this paper for ICLR at this point.

Paper is badly written and the problem it tries to solve is not clearly stated.
Proposed feature embedding is incremental (lack of novelty and technical contribution)
Obtained results are encouraging but not good enough.
Lack of experimental validation.
I think paper can be improved significantly and is not ready for publication at this point.

---

### Decision · Program_Chairs · 2018-01-29
**ICLR 2018 Conference Acceptance Decision**

**Decision:**

Reject

**Comment:**

Three reviewers recommended rejection, and there was no rebuttal.